# Measuring Community Response to Noise—Factors Affecting the Results of Annoyance Surveys [note 1]

**DOI:** 10.3390/ijerph21040420

**Published:** 2024-03-29

**Authors:** Truls Gjestland

**Affiliations:** SINTEF Digital, N-7465 Trondheim, Norway; truls.gjestland@sintef.no; Tel.: +47-932-05-516

**Keywords:** annoyance surveys, exposure–response functions, response scales

## Abstract

Social surveys are conducted to determine how annoyed people are in a certain noise situation. The results are typically presented as exposure–response curves showing the percentage of the area population that are highly annoyed as a function of the noise exposure level. It is a well-known fact that the survey results are not only dependent on the accumulated noise exposure, DNL, DENL, or similar, but also on various other factors such as maximum levels, exposure patterns, noise spectra, etc. A re-analysis of previously reported surveys shows that the results are also, to a large extent, dependent on survey-specific factors like the wording of the annoyance questions, how the questionnaires are presented, response scales, methods of scoring highly annoyed, etc. This paper discusses and quantifies the influence of such factors and suggests ways of comparing results from surveys that have been conducted according to different protocols and different analysis methods.

## 1. Introduction

People who are exposed to environmental noises are affected in various ways. The most prominent effect, i.e., the effect that is experienced by the largest number of people, is annoyance, a concept that, according to Koelega [1], is associated with *disturbance, aggravation, dissatisfaction, concern, bother, displeasure, harassment, irritation, nuisance, vexation, exasperation, discomfort, uneasiness, distress, and hate.* Even with a lack of a more precise definition, annoyance is widely used to characterize the negative impact of noise. The prevalence of highly annoyed residents in a community is used as a measure to quantify the negative effects. Noise regulations are based on a percentage of the area population being highly annoyed, and the noise situation is considered unhealthy if the percentage of highly annoyed residents exceeds a specific limit. Yet there is neither any universally accepted definition of high annoyance, nor any standardized way of measuring it.

Schultz published his synthesis of social surveys on noise annoyance in 1978 [2]. In this paper, he introduced the concept of percentage highly annoyed to quantify the prevalence of annoyance in a community. Various response scales had been used in the surveys that were reviewed. Schultz defined highly annoyed as a response corresponding to the two upper categories of a 7-point numerical scale or the three upper categories of an 11-point numerical scale. This definition represents the upper 27–29 percent of the annoyance scale. Schultz chose a relatively high degree of annoyance in order not to trivialize the annoyance concept. He wanted to include only those for which noise was a serious issue, and not just annoyed persons in general [3]. If a verbal scale was being used, Schultz included those that indicated they were very or extremely annoyed (or using similar modifiers). This method would typically include the upper two categories of a 5-point verbal scale [4].

Later, the US Federal Interagency Committee on Noise, FICON, declared “Annoyance is its preferred summary measure of the general adverse reaction of people to noise, and that the percentage of the area population characterized as “highly annoyed” by long-term exposure to noise is its preferred measure of annoyance” [5]. Since then, this has become a de facto standardized way of presenting the results from social surveys on noise annoyance: the results are shown as so-called dose–response curves, also known as exposure–response functions, ERFs, showing the percentage of highly annoyed residents as a function of the noise exposure.

A lot of work has been concentrated on finding ways to describe the noise exposure in detail, either by direct measurements or predictions, but there has been little concern about the quantity percentage highly annoyed. Therefore, how annoyed, actually, is a person that is highly annoyed, and how is the degree of annoyance determined?

Accurate exposure–response functions are instrumental for regulatory purposes. Exposure limits for environmental noises are often determined on the basis of a certain percentage of the area population being highly annoyed. The European Regional Office of the World Health Organization (WHO), for instance, strongly recommends that noise should be kept below levels corresponding to 10% highly annoyed, as, according to WHO, *noise above this level is associated with adverse health effects* [6]. However, WHO makes no attempt at defining the quantity “highly annoyed”, nor does it give any advice on how the quantity should be determined.

The ISO Technical Specification ISO/TS 15666 [7] gives a recommendation on how social or socio-acoustic surveys should be conducted, but the first version of the document did not define *highly annoyed* at all. Researchers, therefore, relied on the initial ICBEN (International Commission on Biological Effects of Noise) definition [4], which referred back to the original Schultz paper from 1978, or they used their own definition. It was only in the revised 2021 version of the ISO technical specification that a definition of highly annoyed was introduced [7]. But, still, there are a number of variables that need to be taken into account.

The ISO technical specification 15666 introduces two different definitions of high annoyance depending on which response scales, verbal or numerical, have been used in the survey. The TS also recognizes that these two quantities are different and suggests a way of recalculating the verbal score to be comparable with the numerical score. This indirectly indicates that the numerical response scale is the preferred one, but, nevertheless, the quantity percentage highly annoyed is not uniquely defined.

Social surveys on noise annoyance are conducted by many researchers in many countries. New dose–response curves are compared, and differences are discussed with reference to previously reported data and commonly accepted reference curves. However, few researchers seem to realize that most, if not all, of the observed differences are not caused by actual changes in the basic annoyance response, but reflect differences in survey design, questionnaires, analysis methods, etc.

This paper discusses different factors that affect the results of an annoyance survey and suggests ways to compare the results from surveys that have been conducted according to different protocols.

Exposure–response curves typically show the percentage of the exposed population that is highly annoyed as a function of the equivalent noise level, or a derivative such as DNL or DENL. All factors that affect the exposure–response function, other than the accumulated noise level, are referred to as non-acoustic factors. Such factors may modify the annoyance response. This is the case, for instance, for operational changes where people who are exposed to abrupt changes typically are more annoyed. Other factors may affect the apparent results of a survey, but not necessarily the annoyance response itself. The use of different response scales, for instance, may indicate a varying prevalence of annoyance, which is only caused by methodological differences and not by differences in the actual subjective annoyance response.

## 2. Method

### 2.1. Data Collection

Data from previously reported surveys on environmental noise annoyance were collected from the literature. A database compiled by Jim Fields comprising copies of more than 1300 journal articles and reports on surveys on noise annoyance proved to be most valuable. This database covers the period 1960–2008. Data from more recent surveys was found from searches in relevant scientific journals and conference proceedings. The Socio-Acoustic Survey Data Archive (SASDA), established by the Institute of Noise Control Engineering, Japan, was also a valuable source for survey data [8]. Some datasets were provided directly by the researchers responsible for the surveys. All the analyses were based on already completed surveys, and no new surveys were conducted or initiated.

A complete table of all the survey data would be outside the scope of this paper, but the interested reader may find comprehensive tables with relevant survey data in the following research papers:Fidell et al.: A first-principles model for estimating the prevalence of annoyance with aircraft noise exposure [9], (aircraft noise).Gelderblom et al.: On the stability of community tolerance to aircraft noise [10].Gjestland: Recent World Health Organization regulatory recommendations not supported by existing evidence [11], (aircraft noise).Schomer et al.: Role of community tolerance level (CTL) in predicting the prevalence of annoyance from road and rail noise [12].Gjestland: On the temporal stability of people’s annoyance with road traffic noise [13].Fidell et al.: Updating a dosage–effect relationship for the prevalence of annoyance due to general transportation noise [14].Brink: A survey on exposure–response relationships for road, rail, and aircraft noise annoyance [15].Yokoshima et al.: Representative exposure–annoyance relationships due to transportation noises in Japan [16].Miedema et al.: Exposure–response relationships for transportation noise [17].Guski et al.: WHO environmental noise guidelines for the European region: a systematic review on environmental noise and annoyance [18].

### 2.2. The CTL Method

Until recently, the favored way of developing exposure–response functions, ERFs, from annoyance survey data has been by conventional polynomial regression techniques. This method yields curves where two variables, slope and intersect, have been determined. A direct comparison of two curves with different slopes is not trivial.

There is, however, another standardized method to establish such ERFs that facilitates a comparison of different curves. This method, based on the community tolerance level (CTL), is described in the standards ISO 1996-1 [19], and ANSI S 12.9 [20].

The CTL method is based on the assumption that annoyance increases with the noise level at the same rate as the loudness function. This implies that all exposure–response functions have the same “shape”, and the only variable is the positioning of the exposure–response curve relative to the noise axis (*x*-axis). Therefore, as opposed to standard polynomial regression where the objective is to find a function that has “the best fit” to a set of data points, the CTL method seeks to position a fixed function to these data points. The position is described by the noise level at which 50% of the exposed residents are highly annoyed (and the other half not highly annoyed). This is the so-called community tolerance level, CTL or *L*_ct_. A high CTL value characterizes a community that is very tolerant to noise, and, hence, the annoyance with noise is low, and a low CTL value indicates the opposite, low tolerance to noise and a high prevalence of annoyance.

The exposure–response function, i.e., the probability of being highly annoyed at a certain noise level, is given by the following equation:$$p_{HA}=100\;e^{-a}$$
where the exponent *a* is given by: ($\frac{1}{10^{0.1(Lden\;-\;L_{ct}\;+\;4.7)}}$)^0.3^

The complete exposure–response function is uniquely described by a single decibel quantity *L*_ct_. This quantity shifts the position of the ERF back and forth along the noise axis. The difference in the annoyance response between two noise situations can, therefore, be described with a single number, the difference between the CTL values. This quantity can be explained as the number of decibels the noise in one situation must be changed in order to get the same annoyance response as in the other situation.

If community A is characterized by *L*_ct_ = 74 dB and community B by *L*_ct_ = 78 dB, one may argue that residents of community B on average will tolerate 4 dB higher noise levels in order to express the same degree of annoyance as residents of community A.

### 2.3. Data Analyses

New exposure–response functions based on the CTL method for all the selected surveys were established on the basis of reported paired observations of noise exposure and prevalence of high annoyance. For some surveys, we had access to the original individual responses, but, in most cases, we had to rely on pooled data: prevalence of high annoyance per exposure bin, typically 5 dB wide.

The surveys were then sorted and analyzed based on different criteria such as wording of the questionnaire, presentation mode, response scales, principal noise source, etc. This was carried out by the author together with other colleagues. Some of the results from these analyses have been published as separate papers elsewhere [9,10,12,13,21,22]. This paper summarizes the results and compiles them in a way that renders them suitable for direct application when the task is to compare results from social surveys on noise annoyance that have been conducted in different ways and according to different protocols.

## 3. Non-Acoustic Factors

### 3.1. Response Scales

The technical specification ISO/TS 15 666 recommends two standardized questions to be included in a survey [7]. Both deal with the assessment of long-term noise annoyance for a specified period of time (e.g., 12 months). One question refers to a 5-point verbal response scale and the other to an 11-point numerical scale. Highly annoyed is defined by the two upper categories of the verbal scale and the three upper categories of the numerical scale. The document underlines that these two definitions of highly annoyed do not yield identical answers. When reporting survey results, it is, therefore, necessary to specify how the quantity highly annoyed was derived; HA_V_ for the verbal scale and HA_N_ for the numerical scale. HA_V_ is normally larger. The technical specification also has a procedure for transforming HA_V_ to a quantity, HA_VW_ (highly annoyed, weighted verbal response), that can be readily compared with the numerical response.

Gjestland and Morinaga have shown that the average difference between the two quantities is equivalent to a 6 dB shift in the noise exposure [22]. They analyzed 43 annoyance surveys on transportation noise, aircraft, rail, and road traffic, comprising nearly 27,000 respondents in which the participants were asked to assess the noise situation using both a numerical and a verbal scale as recommended by ICBEN [4]. These surveys had been conducted in Germany, Japan, Norway, Switzerland, and Vietnam. References to all the original response data for these 43 surveys can be found in Gjestland and Morinaga [22].

Figure 1 illustrates the effect of making adjustments to the annoyance assessed on the verbal scale. The CTL value based on the numerical scale is normally larger than the CTL value for the verbal scale as shown in panel A (a high CTL value means low annoyance). After an adjustment, the CTL values are brought within a difference of typically less than ±2 dB as shown in panel B.

Since the annoyance assessments in these surveys were carried out by the same individuals and the only outcome of the analysis was the relative difference between the two responses, any possible confounding factors become irrelevant. Gjestland and Morinaga [22] found that people who were classified as highly annoyed according to their verbal responses seemed to tolerate, on average, 6 dB higher noise levels in order for their numerical responses to indicate the same degree of high annoyance.

### 3.2. Mode of Presentation

The earliest surveys were usually conducted as face-to-face interviews in the respondent’s home. Later on, telephone interviews became a favorite method, being faster and less expensive to carry out. Postal surveys are also being used. The potential respondents are contacted by mail or otherwise, and are requested to complete a self-administered written questionnaire, which, in turn, is returned by mail. In some countries, this has become the favored survey method as there is an increasing number of potential respondents that decline to participate in telephone surveys.

The US Federal Aviation Administration recently conducted a large survey by mail [23], but the researchers that were responsible for the survey also complemented the study with a smaller number of telephone interviews with the same mail respondents to check the influence of the survey mode. The participants in these studies, via mail or via telephone, were selected randomly using the same selection protocol. A total of 10,000 individual responses were collected via mail and about 2000 responses via phone interviews. Miller et al. who conducted the survey [23] found that people responding to the written questionnaire, on average, seemed to be more annoyed than people responding to the telephone interview. They found that the difference increased with increasing noise exposure levels. The average difference across the range 50 dB < *L*_dn_ < 75 dB was equivalent to a shift in the exposure of about 5 dB. The results reported by Miller et al. are shown in Figure 2. The two curves represent a logistic fit to the responses from the mail survey and phone survey respectively.

A CTL analysis of the two response categories using data reported in [23] confirmed this finding, the difference in CTL values being 4.8 dB.

The issue of the survey mode is an ongoing discussion. It has, for a long time, been recognized that the response to a survey question may depend on how the question is presented: one-on-one interview, postal or web-based questionnaire, etc.—see, inter alia, Brink [24], Canturia et al. [25], and National Academies of Sciences [26]. The results presented by Miller et al. [23] quantify such differences.

Fidell et al. have analyzed 45 surveys on aircraft noise conducted either face-to-face, via telephone, or as a postal survey [27]. They found no significant differences between face-to-face and telephone interviews, both involving contact with a live agent. However, mail surveys produced a higher prevalence of highly annoyed respondents with an average difference equivalent to a 10 dB shift in the noise exposure. This difference may also be due to the combined effect of other non-acoustic factors, but the result shows the same tendency as observed by Miller et al. [23]. More survey results that allow a direct comparison of the two survey modes—interview by a live agent or a self-reporting written questionnaire—as reported by Miller et al. have not been found. Based on the other analyses reported above, however, we find it plausible that the difference between the two modes corresponds to a shift in the noise level of at least 5 dB, and people dealing with a live survey agent report lower annoyance.

### 3.3. Changes in Airport Operations

The effect of abrupt changes in the airport operations has been observed in many aircraft noise studies. Most airports experience a gradual increase in traffic over the years. In most cases, this growth is small, and week-to-week or year-to-year changes in the noise exposure will hardly be noticed by the neighborhood community. However, occasionally, abrupt changes will occur such as the opening of a new runway, the introduction of a new fleet of aircraft (for instance, if a major airline is moving to a new hub), the introduction of new operational procedures and new flight trajectories, etc.

Janssen and Guski [28] have presented a study on temporal trends in the aircraft noise annoyance response. They analyzed a set of 32 aircraft noise studies contained in the TNO database. They observed that abrupt changes in the airport operations will affect the annoyance response, and, therefore, introduced a classification procedure as follows:


*We call airports “low-rate change airports” (LRC), as long as there is no indication of a sustained abrupt change of aircraft movements, or the published intention of the airport to change the number of movements within 3 years before and after the study. An abrupt change is defined here as a significant deviation in the trend of aircraft movements from the trend typical for the airport. Each trend is calculated by means of total movement data during a five-year period. If the typical trend is disrupted significantly and permanent, we call this a “high-rate change airport” (HRC). We also classify an airport in the latter category if there has been widespread public discussion about operational plans within 3 years before and after the study.*


Janssen and Guski found that the average difference in the annoyance response between an HRC and an LRC airport was equal to a 6 dB shift in the exposure level.

Gelderblom et al. [10] have made a similar analysis of a set of 62 aircraft noise annoyance surveys to study the stability of community tolerance to noise. Their dataset comprised about 650 paired observations of aircraft noise exposure and the prevalence of high annoyance. Gelderblom et al. carried out a classification of the airports according to the protocol suggested by Janssen and Guski and defined 45 LRC airports and 17 HRC airports. They found that the average difference in the annoyance response between these two categories of airports was equal to a 9 dB shift in noise exposure. People living near a high-rate change airport seem to tolerate 9 dB less noise in order to express the same degree of annoyance as residents living in a low-rate change airport community.

The results reported by Gelderblom et al. [10] are shown in Figure 3. The figure shows CTL values for aircraft noise surveys conducted over a period of about 35 years. The airports have been classified as LRC (blue triangles) or HRC (red squares) according to the proposed method by Janssen and Guski. Trendlines have been fitted to the two datasets. Both trendlines have a small positive slope, indicating that the CTL values increase a little for more recent studies. This indicates a decrease in the annoyance response, contrary to the claim by several authors [18,29]. The average difference between the two lines is about 9 dB.

The increased prevalence of annoyance due to a situation that makes the airport an HRC airport actually seems to last longer than the three-year period suggested by Janssen and Guski. This is discussed by Gelderblom et al. [10]. They argue that the shift in a person’s attitude may be permanent, and that the community response only shifts gradually as people move in and out of the airport neighborhood.

Many new noise surveys are conducted because noise has become an issue of public debate, and the residents demand proof of the actual annoyance situation. Since surveys are time-consuming and expensive to carry out, it is more likely than not that new study sites are located at an HRC airport rather than at an LRC airport. A greater portion of HRC airports in the total number of noise surveys may explain the claim that aircraft noise annoyance is increasing [30]. For most of the airports characterized as HRC in this study, the classification was based on quite recent operational changes or recent announcements of controversial plans.

We find it plausible that a similar increase in the annoyance response due to abrupt changes may also be observed for other noise sources, for instance, in connection with the construction of a new road or a major refurbishing of an existing one.

### 3.4. Traffic Volume

Annoyance with transportation noise increases when the traffic volume increases. An increased number of noise events per day will increase the DNL level. However, the annoyance seems to increase at a faster rate than the equivalent level.

Gjestland et al. [21] have studied the prevalence of noise-induced annoyance and its dependency on the number of aircraft movements. They analyzed the results from 32 aircraft noise surveys and concluded that, for a given noise exposure level, the percentage of highly annoyed residents increased equivalent to a DNL increase of 1.8 dB per doubling (= 6 × log(2)) of the number of aircraft movements. This increase comes in addition to the regular 3 dB per doubling.

Consequently, residents living near a small airport seem to tolerate 6 dB more noise than neighbors to an airport with ten times more traffic (= 6 × log(10)) in order to express the same degree of annoyance.

A similar analysis of road traffic noise surveys shows the same tendency, but with a slightly smaller effect: 1.5 dB per doubling (= 5 × log(2)) [31].

## 4. Reference Curves

The exposure–response curves for transportation noise sources developed by Miedema and Vos [17] are widely used as standard references. These so-called Miedema curves are simple second-order polynomial regression functions fitted to the results of a selection of surveys. The curves were refined by Miedema and Oudshoorn [32] who used a third-order regression function. The Miedema curve for aircraft noise is almost identical to a CTL function anchored at *L*_ct_ = 73.3 dB. An analysis of the input data used by Miedema and Vos when they derived at this average ERF for aircraft noise shows that the curve can be used to predict the dose–response function for an airport with about 250,000 aircraft movements per year. For airports with other traffic volumes, the curve should be adjusted according to paragraph 2.7.

Schomer et al. [12] have shown that a CTL function anchored at *L*_ct_ = 78.3 dB is a good approximation for the Miedema curve for road traffic noise. This curve seems to fit a situation with a traffic volume of about 2000 ADT (annual daily traffic) [31].

Figure 4 shows the Miedema curves for air, road, and rail traffic (solid lines), together with the CTL curves (dashed lines) that are fitted to similar datasets.

The difference between the Midema curves and the CTL functions seem to increase towards higher noise levels. This is mainly due to the fact that the Miedema curves are polynomial regression functions, and the CTL curves are logistic functions.

The Miedema curves for transportation noise are based on surveys conducted between 1965 and 1994. They can still be considered valid according to the findings by Gelderblom et al.

## 5. Comparison of Survey Results—Three Examples

### 5.1. The US Environmental Survey, NES

The results from the recent 20 US airports study [23] were used to construct a new US national average dose–response curve. At first glance, this curve indicated a much higher prevalence of annoyance than the earlier FICON curve [14] and the Miedema curve. The average CTL value for the responses at 20 airports was *L*_ct_ = 60.2 dB. For comparison, the Miedema curve for aircraft noise has a CTL value *L*_ct_ = 73.3 dB (see Figure 4). Miller et al. concluded that annoyance with aircraft noise had increased markedly. A closer look at the Miller et al. report, however, reveals that the two curves cannot be directly compared. The surveys that were the basis for the Miedema curve were conducted as face-to-face or telephone interviews, whereas the 20-airport study was a mail survey (only the mail responses were used.) Miller et al. have showed in their report that the difference in the response mode for this particular study was equal to a 5 dB shift in the exposure (see Figure 2).

Similarly, the scoring of high annoyance carried out by Miedema and Oudshoorn was based on a cut-off of around 72% on the annoyance scale. Miller et al. defined high annoyance as the two upper categories of a 5-point verbal scale, equivalent to a 60% cut-off. The individual verbal responses of the 20-airport study have been re-calculated as described in the standard ISO 15666:2021 with category #5 counted in full and category #4 given a weight of 0.4. The difference between the two procedures for scoring the prevalence of high annoyance turned out to be equal to a 5 dB shift in the noise exposure. This is a little less than the average value found by Gjestland and Morinaga [22]. The average traffic volume at the 20 US airports was about 280,000 movements per year, so the required adjustment for traffic volume when comparing with the Miedema curve is only 0.3 dB (= 6 × log(280,000/250,000)).

Therefore, the reported exposure–response curve for the 20-airport study, described by the CTL value *L*_ct_ = 60.2 dB, should be adjusted by 5 dB to account for mail vs. live agent mode, and an additional 5 dB for the definition of high annoyance, plus 0.3 dB for the traffic volume. The new value *L*_ct_ = 70.5 dB (60.2 + 5 + 5 + 0.3) can be compared with the Miedema curve. This is shown in Figure 5. The two curves are quite similar for exposure levels below about *L*_DN_ = 60 dB, the range of primary interest for regulatory purposes. This indicates that reactions to aircraft noise at low exposure levels in the US today is similar to what was found by Miedema and Vos 25 years ago. The large difference between the two curves at high exposure levels should not be considered problematic and is probably caused by methodological differences in the curve-fitting process. Exposure levels that leave 40–50% of the population highly annoyed are not acceptable anyway.

### 5.2. Japanese Road Traffic Noise Surveys

Yokoshima et al. have reported ten surveys on road traffic noise conducted between 1994 and 2011 [16]. Nine of the Japanese surveys have similar results, having CTL values between *L*_ct_ = 71 dB and *L*_ct_ = 81 dB, whereas one survey from Kanagawa (SASDA reference JPN010RT1999), reported by Yokoshima and Tamura [33], is a typical outlier with *L*_ct_ = 54.6 dB. This study was excluded from the analysis by Yokoshima et al. A CTL function, *L*_ct_ = 72.3 dB, was fitted to the accumulated data reported by Yokoshima et al. They reported an average exposure–response curve for road traffic noise in Japan to be somewhat above the Miedema curve. However, in the Japanese surveys, the respondents were given a written questionnaire to be filled in and returned by mail (or collected otherwise). For a direct comparison with the Miedema curve, the response should, therefore, be adjusted between 5 dB and 10 dB, as explained in paragraph 3.2.

The resulting adjusted average exposure–response curve for road traffic noise in Japan is quite similar to the Miedema curve as shown in Figure 6 (minimum adjustment: 5 dB), indicating that, at low exposure levels, people in Japan seem to be somewhat less annoyed by road traffic noise than predicted by the Miedema curve.

### 5.3. Norwegian Average ERF for Aircraft Noise

Gelderblom et al. [34] reported results from surveys at five Norwegian airports. These airports can be considered representative for Norwegian airports, serving mainly intermediate-range aircraft like B737 and A320. The largest airport, OSL, has about five times more movements per year than the smallest one, TOS.

For direct comparison with the Miedema curve, the CTL values must be adjusted for the traffic volumes. In addition, Oslo airport, OSL, is a typical HRC airport. The OSL result, therefore, needs to be adjusted between 6 dB and 9 dB, as explained in Section 3.3. Here, we have chosen the mean value, 7.5 dB. The adjustments and results are listed in Table 1.

The average adjusted CTL value for these five airports is *L*_ct_ = 75.2 dB ±1.7 dB. The results are quite similar for all five airports, indicating that the response to aircraft noise is stable across all of Norway.

This average CTL value for Norway can be compared directly with the CTL value for the Miedema curve *L*_ct_ = 73.3 dB, which is slightly lower. In other words, the average annoyance response for aircraft noise in Norway indicate that people in Norway “tolerate” about 2 dB higher noise levels in order to express a certain degree of annoyance, than predicted by the standard reference curve. The corresponding exposure–response curves are shown in Figure 7.

## 6. Conclusions

A review of previously conducted surveys reveals that the results are not only dependent on acoustic and community-specific parameters, but also on survey protocols and how the results are analyzed. The traditional way of comparing results from social surveys is to compare exposure–response functions.

The Miedema and Vos reference curves were constructed on the basis of surveys conducted as face-to-face or telephone interviews, and the scoring of high annoyance was based on a 72% cut-off of the annoyance scale. For comparison purposes, new exposure–response curves should either be constructed on the same basis, or they should be adjusted as explained in this paper depending on the objectives of the comparison. A detailed analysis of many previous surveys shows that what has been presented as “new” results, usually an increased prevalence of annoyance, most often are caused by differences in survey design and analysis methods rather than actual differences in the annoyance response.

There are few indications that annoyance caused by transportation noise has changed significantly over the past 50 years.

## Figures and Tables

**Figure 1 ijerph-21-00420-f001:**
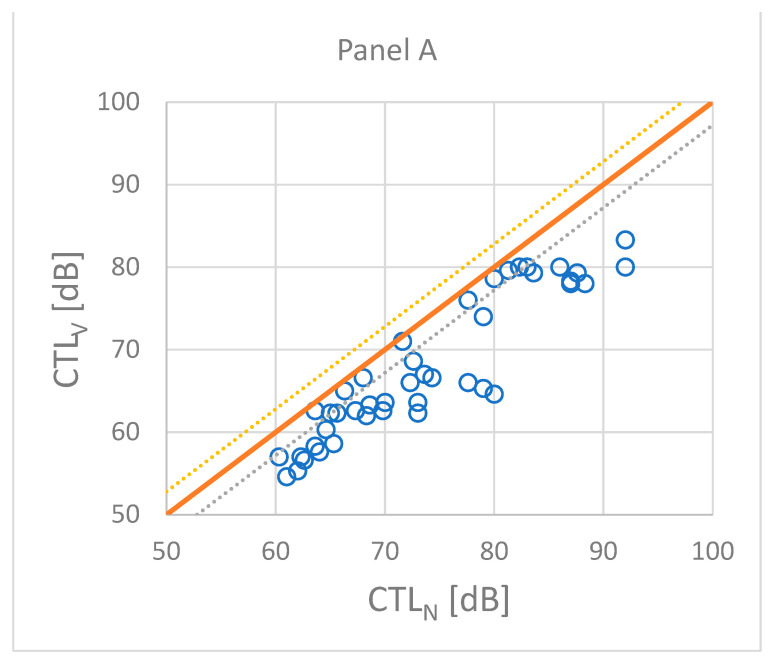
Comparison between CTL values calculated from the response on a numerical scale and from the response on a verbal scale (**Panel A**) and a weighted response on a verbal scale (**Panel B**) as described in ISO/TS 15666. The dashed lines indicate an interval of ±2 dB. Panel A shows that all CTL_V_ values are smaller than the corresponding CTL_N_ values, thus demonstrating that the verbal response always indicates a greater prevalence of high annoyance than the numerical response.

**Figure 2 ijerph-21-00420-f002:**
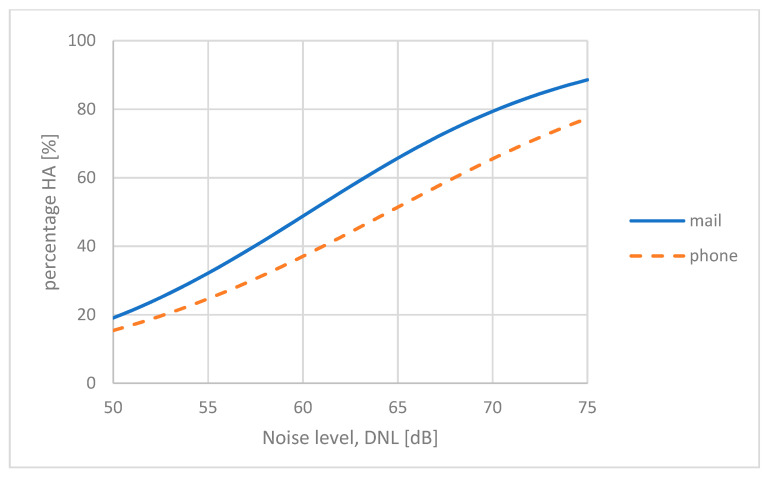
Results from a survey on aircraft noise annoyance conducted either by mail or as telephone interviews from the same survey respondents [23].

**Figure 3 ijerph-21-00420-f003:**
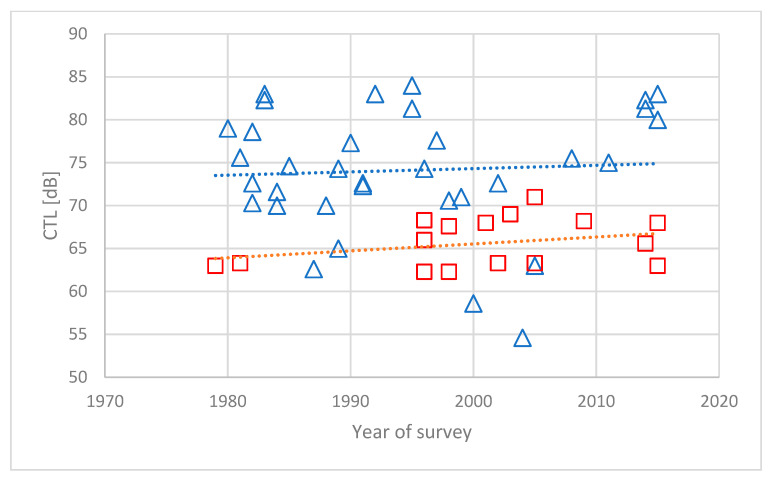
CTL values for 51 surveys on aircraft noise annoyance conducted after 1975. Blue triangles indicate LRC airports and red squares HRC airports (see text for explanation). Trendlines (dashed) for the two datasets indicate growing CTL values corresponding to a decrease in the prevalence of annoyance [10].

**Figure 4 ijerph-21-00420-f004:**
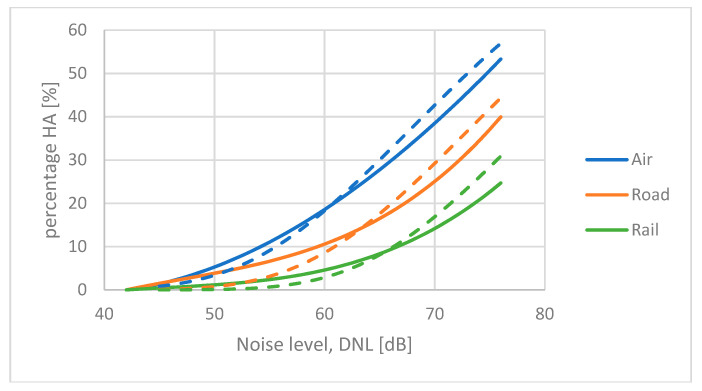
Exposure–response curves for transportation noise. Solid lines are third-order polynomial functions developed by Miedema and Oudshoorn [32]. Dashed lines are CTL functions for *L*_ct_ = 73.3 dB (air), *L*_ct_ = 78.3 dB (road), and *L*_ct_ = 83.5 dB (rail).

**Figure 5 ijerph-21-00420-f005:**
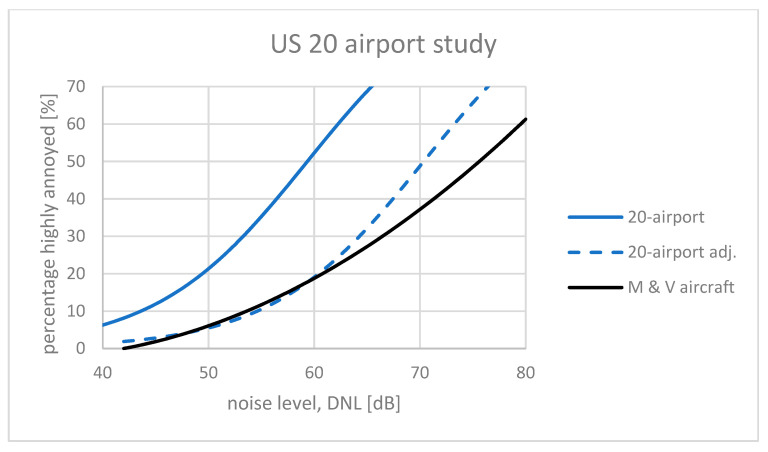
Exposure–response curves for the US 20 airport study adjusted according to response scale, traffic volume, and mode of survey presentation (see text).

**Figure 6 ijerph-21-00420-f006:**
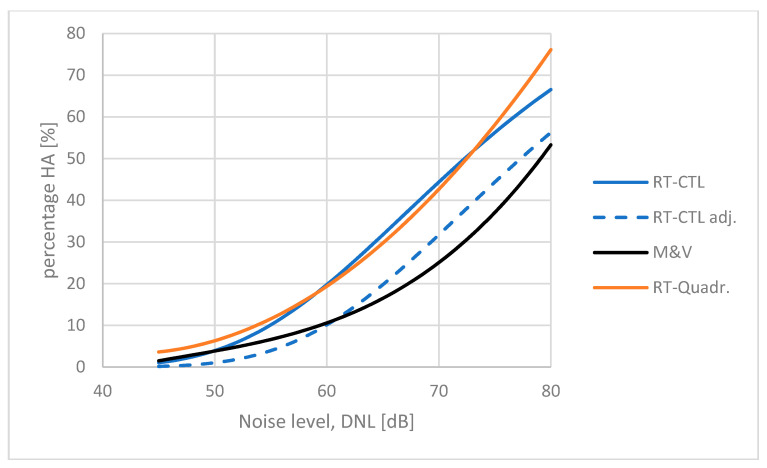
Exposure–response curves for road traffic in Japan. Quadratic regression function reported by Yokoshima et al. [16] (red solid line). Corresponding CTL function: *L*_ct_ = 72.3 dB (solid blue line). CTL function adjusted 5 dB for postal vs telephone interview (dashed blue line). Miedema curve for road traffic noise (solid black line).

**Figure 7 ijerph-21-00420-f007:**
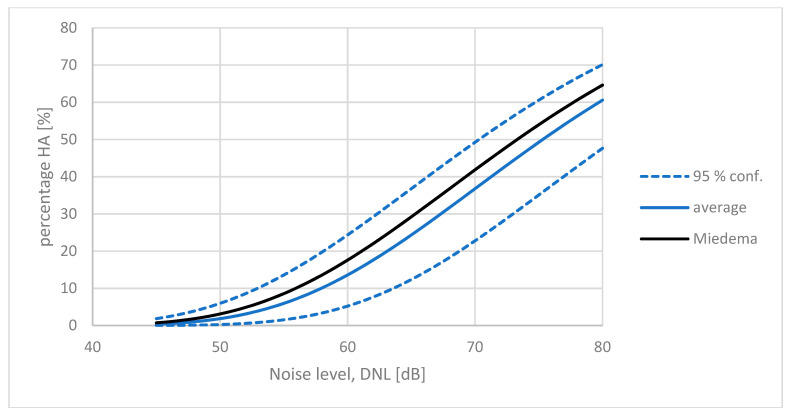
The average adjusted exposure–response curve for five Norwegian airports (solid blue line) compared with the reference curve by Miedema and Vos (black line): *L*_ct_ = 73.3 dB. The dashed lines indicate the outer 95% confidence interval for all five individual ERFs.

**Table 1 ijerph-21-00420-t001:** Data for surveys at five Norwegian airports.

Airport	Movements	CTL	−95% Conf.	+95% Conf.	6 × log n	HRC LRC	CTL Adjusted
OSL	250,000	67.0	−2.1	2.0	0.0	7.5	74.5
SVG	94,000	77.7	−2.4	2.5	−2.5	0	75.2
TRD	65,000	79.3	−2.8	3.1	−3.5	0	75.8
BOO	55,000	76.6	−2.4	2.3	−3.9	0	72.7
TOS	47,000	82.2	−2.8	3.2	−4.4	0	77.8

Airports identified by their IATA code. +/− 95 % confidence interval. HRC/LRC high/low rate of change as explained in the text.

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
