# Peer review of "Measuring Community Response to Noise—Factors Affecting the Results of Annoyance Surveys†"

_ijerph, 2024, doi:10.3390/ijerph21040420_

Round 1

Reviewer 1 Report

Comments and Suggestions for Authors

This article describes the factors that influence the results of surveys on noise annoyance. The author takes a critical look at these factors and uses examples to find explanations for deviations between studies, when these can be quantified. This is an important contribution to avoid misinterpretations when comparing ERCs. The summarized factors also show the extent of the influencing factors and their relevance.

 The data collection carried out for this purpose under 2.1. lists the referred sources. This is done with an exemplary list of relevant literature. The standardized CTL method is also described and applied.

The data analysis based on the CTL method described in section 2.3. summarizes the results of the used studies. At this point, please summarize the most important values such as the number of studies and respondents. If possible, the effect measures of the studies should be given. A table would probably be suitable for this purpose.

 In connection with the influence of the response scales 2.4, it is recommended to examine and discuss the results of the article “Pooling and Comparing Noise Annoyance Scores and “High Annoyance” (HA) Responses on the 5-Point and 11-Point Scales: Principles and Practical Advice, Brink et al. ijerph 2021

 An outline of the influence of non-acoustic factors should also be provided. These also provide explanations for the different noise responses and for the CTL (e.g. Maarten Kroesen Testing a theory of aircraft noise annoyance: A structural equation analysis, Acoust. Soc. Am. 123, 4250–4260 (2008).

 Although the influencing parameters are clearly described in the individual sections, it would be helpful if these were summarized once again. A table could be used for this purpose. It would be even better if a chart showed the individual contributions of the factors and the overall influence. A separate contribution could also be made for the non-acoustic factors, even if the proportion cannot be quantified. 

In the conclusions, an absolute instruction is formulated: The curves for comparison purposes with reference curves must either be created using the same methods or must be adjusted according to the rules of the submitted article. This strong formulation is not necessarily to be followed. An excessive degree of adjustment and the transfer of results from other studies can eliminate the peculiarities of specific settings. This is particularly the case when non-acoustic factors have a relevant influence on response behaviour. The wording should be more differentiated and toned down. The adjustment depends on the research question to be answered in a particular study, this should be taken into account.

The last sentence of the conclusions is not supported by the reported data of this article.

Reviewer 2 Report

Comments and Suggestions for Authors

Dear Editor, dear author;

Before beginning the analysis of the article I have to emphasize that the interest of the contribution is high. The impact of environmental noise on individuals residing near noise sources will continue to be a hot topic in the coming years. The author makes a review of how to account for variability in findings across studies that try to asses the exposure-response of communities to environmental noise. I base my assessment of the importance of the topic on three examples

Example 1, Meta-analyses should try to follow a systematic and transparent process to statistically combine data from individual studies and evaluate the overall quality of evidence on the effects of environmental noise on annoyance (but also, sleep disturbance, well-being, quality of life, and other health effects).

Example 2, on the other hand, the interest of various studies is in searching for the cause of the differences between noise exposure-response curves (cultural explanations, building characteristics, propagation issues, social inequalities, etc.).

Example 3. Following the publication of the WHO Guidelines on Environmental Noise for the European Region, new surveys have been published that try to find out if there are significant changes in the evidence on the effects of noise on annoyance established through questionnaires compared to the criteria set out by the WHO and adopted as an amendment of the European Environmental Noise Directive.

In all examples, the researchers need to deal with different methodologies (used in both, social surveys and exposure estimation) to explain part of the variation of results between studies. In this paper, the author reviews the artifacts that could cloud genuine comparisons between such studies. I am sure that this article will be highly appreciated in those studies whose methodology is based on the comparative analysis of environmental noise exposure-response curves.

In summary, a global analysis of the study indicates that:

  • The title of the paper does anticipate what the article is about. Maybe I change the hyphen,  Measuring Community Response to Noise: Factors Affecting the Results of Annoyance Surveys
  • The abstract provides an accurate representation of the article.
  • The justification of the study and the significance of the work is clear since the subject under study is current.
  • The scientific environment in which the research is carried out is correctly introduced, since aspects of the research related to the factors associated with the survey methods and how they affect the final result are introduced profusely.
  • The explanations are easy to follow.
  • The references supporting the introduction are quite current (14 scientific citations were published in the last 10 years).
  • The extension of the manuscript is correct given the contents addressed.

After the reading of the paper with the code: ijerph-2887459, titled: Title: Measuring Community Response to Noise - Factors Affecting the Results of Annoyance Surveys; my recommendation is to publish it with minor corrections.

From here it will be explained what aspects of the work generate some doubts and and where I suggest changes to try to improve the paper.

These aspects are the three following:

1. THE FORMAL STRUCTURE OF THE ARTICLE. The presentation of the contents is clear, but this paper raises doubts regarding its organization and design. I am going to present these doubts to the author and I trust that he will explain them. At first glance, I assumed that I had a manuscript in my hands that was a systematic review, but in reality, it was not. Maybe it can be defined as "a research notes presenting preliminary results using literature case studies as an examples". Despite this, I wonder if a discussion section should not be included if only to discuss the implications of this work, its limitations, and how it should proceed in future research. So, determining whether and how these preliminary results can be generalized through statistical analysis.

2. OBJECTIVE. There is a difference between the objective of the paper cited at the end of the introduction and the objective stated in the abstract, being the latter more ambitious and written as a scientific objective. I say this because it is not only about discussing but also quantifying the influence of the factors that affect the results of an annoyance survey…

3. INTRODUCTION. Although a terminological discussion about the meaning of "high annoyance" is interesting, it seems too extensive to me.

Comments on the Quality of English Language

Some problems were detected.  Some examples:

Line 8. (this third person singular subject-verb agreement error appears on other lines) … population that are highly annoyed.. “..population that is highly annoyed..”

Line 19 (this type of error appears on other lines). People that are exposed “People who are exposed”

Line 31 ….the concept percentage highly annoyed.. “…the concept of the percentage of highly annoyed …”

Line 90. “… was found from searches in relevant scientific journals. “… was found from searches in relevant scientific journals..”

Line 122 and others. ….exposure response… “… exposure-response…”

lines 127 and 137. Please try to avoid repetitions - “Community Tolerance Level (CTL)”,

Line 168 (this third person singular subject-verb agreement error appears on other lines) …and compiles them in a way that render them suitable…. “…and compiles them in a way that renders them suitable…”

lines 265 to 272. It is not clear to me the need to introduce literal quotes, such as those expressed in this paragraph.

etc.

Reviewer 3 Report

Comments and Suggestions for Authors

1) Paragraphs 2.4 through 2.8 introduce one factor affecting the results of social surveys in each section. Since these factors are different in nature from the CTL method explained in 2.2, and 2.3, which explains the methodology of the study conducted in this paper, we think it would be easier to understand if the chapter structure were changed.

2: L.194) Figure 1 compares before and after correction, but lacks an explanation of the correction method.

3: L.252) The title of the section is "Operation changes", but it is not clear that this is a transportation operation, so please consider changing the title. Also, "This effect" at the beginning of the sentence should identify what the effect is.

4: L.293) Please add a legend to the Figure 3.

5: L.356) Some factors affect all, such as the method of response collection and the scale used, while others, such as the type of transportation system, may require different factors to be considered. In ordr to clarify the correction method proposed in this paper, please outline the correction method at the end of chapter 2 or the beginning of chapter 3. It would be good to have a diagram or something like that.

Comments on the Quality of English Language

"Lct" is not consistent, so please check. If we follow the notation in other publications, it would be preferable to use italics for "L" and lowercase for "ct" as cubic.
